# Individual Feature Selection of Rolling Bearing Impedance Signals for Early Failure Detection

Florian Michael Becker-Dombrowsky *[ID], Quentin Sean Koplin and Eckhard Kirchner [ID]

Department of Mechanical Engineering, Institute for Product Development and Machine Elements, Technical University of Darmstadt, Otto-Berndt-Straße 2, 64287 Darmstadt, Germany; contact@koplin-mail.de (Q.S.K.); kirchner@pmd.tu-darmstadt.de (E.K.)
* Correspondence: florian_michael.becker@tu-darmstadt.de

**Abstract:** Condition monitoring of technical systems has increasing importance for the reduction of downtimes based on unplanned breakdowns. Rolling bearings are a central component of machines because they often support energy-transmitting elements like shafts and spur gears. Bearing damages lead to a high number of machine breakdowns; thus, observing these has the potential to reduce unplanned downtimes. The observation of bearings is challenging since their behavior in operation cannot be investigated directly. A common solution for this task is the measurement of vibration or component temperature, which is able to show an already occurred bearing damage. Measuring the electrical bearing impedance in situ has the ability to gather information about bearing revolution speed and bearing loads. Additionally, measuring the impedance allows for the detection and localization of damages in the bearing, as early research has shown. In this paper, the impedance signal of five fatigue tests is investigated using individual feature selection. Additionally, the feature behavior is analyzed and explained. It is shown that the three different bearing operational time phases can be distinguished via the analysis of impedance signal features. Furthermore, some of the features show a significant change in behavior prior to the occurrence of initial damages before the vibration signals of the test rig vary from a normal state.

**Keywords:** condition monitoring; rolling bearing; feature engineering; damage early detection; electrical impedance measurement

## 1. Introduction

Fault-based breakdowns of rotating machinery reduce the reliability, security, and availability of machines [1]. Thus, detecting abnormalities becomes more important to reduce unplanned downtimes. Rolling bearings are one of the most reliable machine elements and are used in a wide range of different rotating machines [2]. They are located in the flux of forces, which means that all changes or harmful abnormalities in rotating machines' behavior interfere with them. Because of that, nearly 20% of all machine failures are based on rolling bearing damages [3]. Monitoring the bearing condition can reduce unplanned downtimes and increase the availability of technical systems. This kind of condition monitoring is the basis for condition-based maintenance or so-called predictive maintenance [2].

The aim of predictive maintenance is to forecast a machine breakdown using condition monitoring and to fulfill necessary maintenance steps at an optimum time slot [2]. A fundamental step for condition monitoring is the detection of failures and the classification of machine element conditions [1,2]. The data acquisition using sensors and sensor systems is essential for different observed parameters to receive information about the monitored system [1].

Early research at the Institute for product development and machine elements shows the applicability of ball bearings as sensors for an in situ load and failure monitoring [4].

This concept uses the electric properties of rolling bearings to calculate the bearing load and gather information about the bearings' condition and operational state [4]. Martin et al. show that the electric impedance signal changes over the lifespan of a ball bearing, and three different phases in the bearing life are distinguished. The occurrence of surface damage is observed in the real and imaginary parts of the impedance signal. Furthermore, it is possible to localize the damage and measure its length by analyzing the impedance data using the characteristic ball-bearing frequencies [5,6]. Maruyama et al. show that measuring the impedance can monitor the lubrication condition [7]. All this displays the opportunities of the electric impedance measurement for ball bearings, which are further investigated in this paper by analyzing the impedance signal itself and features calculated from it to describe the rolling bearing life span.

### 1.1. Condition Monitoring Using Vibration Data

A common solution for condition monitoring in rolling bearings is measuring the vibration signals resulting from normal and abnormal behavior of the observed components. In the case of pitting, vibrations occur when the damage already harms the surfaces of the contact partners. Overrolling surface damage in the bearing runways or rolling elements leads to a pulse excitation, which is intensified by the elastic material behavior of the components. The resulting vibrations are transferred to the sensor through the structural components, where these are detected and sent to analyzing systems. The signals are prepared for further investigations in the time domain, frequency domain, and time-frequency domain. These data are the basis for receiving information about the system condition and prediction of the remaining operational time [1–3].

In order to predict the remaining life of the bearing, machine learning methods like feature engineering and regression models are used [1,8,9]. The sensors providing the necessary data are not located directly at the monitored component, which is why their signals are a combination of source effects like damages in the bearing runway and transmission path effects influenced by the structural components and their interference [1–3]. This can be a disadvantage because the affectation of the signals can be found in the data used for the machine learning techniques, which leads to uncertainties in the models. Therefore, interfering signals have to be minimized by filtering and other mathematical operation [1,8,9], increasing the complexity of the algorithms. Furthermore, information from the point of interest about the condition of the monitored machine elements is missing.

The impedance is frequency-dependent, which is why it can be investigated in the time and frequency domain [5,6,10]. Since many signal features for vibration data in the time and frequency domain are already commonly used for condition monitoring [1,2,8,9], these existing features will be used for individual feature selection as a feature engineering method in this work.

### 1.2. Feature Engineering

A feature is a mathematical quantity that describes the attributes and characteristics of a measurement signal. Features are created to decrease the amount of data and to create robust predictors of a specific characteristic of interest [11]. The process of feature engineering is used to create meaningful features with the highest possible quality of information concerning the desired characteristic. Features quantify certain characteristics of a signal. Prominent examples of such features are the mean value and the standard deviation of a set of measurements. Features derived from the time domain describe the temporal behavior of the measurand. Additional features are derived from the frequency domain. It is, therefore, necessary to calculate the frequency spectrum of the impedance signal by applying a discrete Fourier transform. [3]

Feature engineering involves the following steps: First, the measurement signal is preprocessed. Preprocessing the signal enables the reduction of errors such as background noise and errors in the measurement setup [1]. The resulting signal is used to generate features. This can be accomplished by calculating statistical measures like the standard

deviation of the signal or by using mathematical methods such as Fourier transform before applying mathematical operations. After generating multiple features, the results are compared to each other to find the features with the most significance regarding the desired information or characteristic. Ranking the individual features according to a specified criterion to select the most valuable ones is called individual feature selection. The criterion quantifies the relevance of each feature [12].

Feature engineering is an important part of further signal-analyzing steps. To fulfill tasks like condition classification and early damage detection, features need to be generated and implemented in machine learning algorithms [1–3].

### 1.3. Electric Behavior of Rolling Bearings

In an electric circuit, ball bearings show capacitor-like behavior since the electrically conductive components are separated by electrically isolating lubrication films. Depending on the lubrication film thickness, three different electric behaviors are observed and modeled as an equivalent circuit. These behaviors can be modeled in the following way. The Hertzian contact zone is described as a plate capacitor, which is illustrated in Figure 1, so the capacity in the elastohydrodynamic (EHL) contact can be calculated using the capacitor equation [10]:

$$C_{Hz} = \varepsilon_r \varepsilon_0 \frac{A_{Hz}}{h_0},$$ (1)

where $A_{Hz}$ is the Hertzian contact area, $h_0$ the central lubrication film thickness in the EHL contact, and $\varepsilon_r \varepsilon_0$ the permittivity of the lubricant.

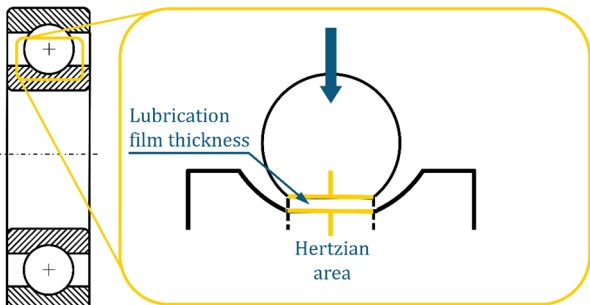

**Figure 1.** Electric model of the EHL contact in a ball bearing [13].

Gemeinder and Barz enhance the model by initiating a factor $k_r$ considering the influence of the border zone [14,15]:

$$C_{Hz} = k_r \varepsilon_r \varepsilon_0 \frac{A_{Hz}}{h_0}.$$ (2)

Schirra shows that the factor $k_r$ is not constant, and Puchtler et al. considered the influence of the unloaded rolling elements in the model [13,16].

The description of the EHL contact as a plate capacitor is only possible when a separating lubrication film exists. In the case of dry friction, direct metallic contact between the rolling elements and the runways leads to a resistive behavior, which means that this condition can be understood as ohmic resistance. An intermediate state can be observed when the lubricant separates the rolling elements and the runway completely so that metallic contacts are avoided. It can be described as an ohmic resistance in parallel connection to a plate capacitor. When the lubrication film is not thick enough, harmful EDM currents occur, damaging the surfaces. This needs to be avoided for sensory usage of the impedance measurement method. Figure 2 gives an overview of the three conditions [10,17–19].

In the case of a sufficiently thick lubrication film, the contact can be modeled as a plate capacitor, whose plate thickness is the lubrication film thickness and whose plate area is the Hertzian area, cf. Figure 1. Film thickness and Hertzian area, and thus also the capacitance,

depend on the load. In this study, the complex impedance is measured, which reflects the entire electric behavior of the bearing, including resistive and capacitive terms. In the case of capacitive behavior, the phase angle tends to $-90°$, which can be used as an indicator for lubrication conditions. For phase angles of about $0°$, an ohmic behavior can be observed, and metallic contacts occur [15,18,19].

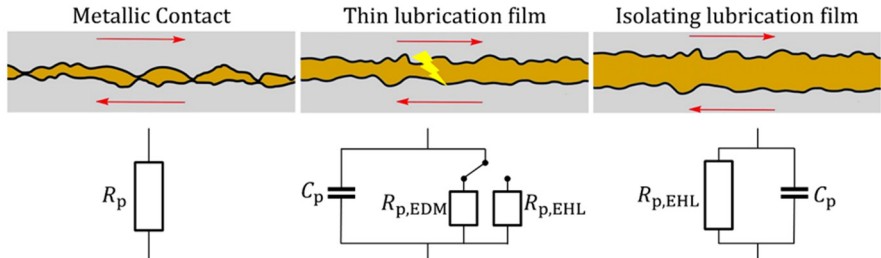

**Figure 2.** Electric model of the EHL contact as a function of the lubrication film thickness [6].

Because of the usage of the electric properties of rolling element bearings, hybrid bearings, or full ceramic bearings applied in, e.g., electric machinery, cannot be observed using the impedance due to the missing electrical conductivity. For these bearings, classic monitoring approaches have to be used and optimized using feature engineering and other techniques [20,21].

*1.4. Research Design*

The aim of this work is the further investigation of rolling bearing impedance data from five fatigue tests generated by Martin et al., which already showed the possibility of impedance measurement for rolling bearing observation [5,6]. In their research, only the parameters listed in Table 1 are analyzed, but no additional features were identified or investigated [5,6]. To further identify and analyze additional features is the aim of this contribution. The identified features will build the fundamentals for explainable machine learning algorithms as part of future research. Because impedance measurement for condition monitoring is a new approach in this field, it must be clarified whether the generated signals are appropriate for use in machine learning algorithms like classifiers. Therefore, the focus of this work is to first investigate the opportunities of impedance-based data for condition detection.

**Table 1.** Signals calculated from the measured complex impedance signal.

| Description | Formula | Unit |
|---|---|---|
| Real part (effective resistance) | $R = Re(\underline{Z})$ | $\Omega$ |
| Imaginary part (reactance) | $X_{LC} = Im(\underline{Z})$ | $\Omega$ |
| Absolute value (apparent resistance) | $Z = \sqrt{R^2 + X_{LC}^2}$ | $\Omega$ |
| Phase angle | $\varphi = \arctan\left(\frac{X_{LC}}{R}\right)$ | rad |

To do so, the impedance signals by Martin et al. are filtered and preprocessed to remove outliers. Based on the state of research for condition monitoring using vibration data, time and frequency domain features are calculated from the impedance data and are analyzed. The suitability of these new features will be checked using a normalized label over the operational lifetime of the rolling bearings. A phenomenological explanation of the feature behavior will be provided afterward. The results of the analysis are compared to a different impedance measurement approach with different types of rolling bearings in a validation fatigue test to obtain an indication of a possible generality of the extracted signal information.

## 2. Materials and Methods

In this section, the used impedance measurement methods and the test rig are presented. After that, the test parameters are introduced.

### 2.1. Impedance Measurement Methods

Martin et al. used a voltage divider to detect the impedance [5]. The equivalent circuit is shown in Figure 3.

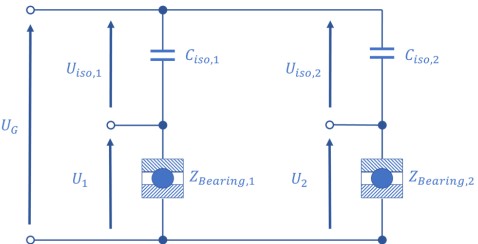

**Figure 3.** Equivalent circuit of the voltage divider for impedance measurement following [5].

The voltage of the generator and the voltage over the reference impedance are detected. Therefore, the capacity of isolation of the test rig is measured and applied as a reference. In the used configuration, two rolling bearings are observed simultaneously. The impedance is calculated from the known capacity and the measured voltages using the following equations [6]:

$$Z_{Bearing,1} = \left( \frac{U_G}{U_{iso,\,1}} - 1 \right) \cdot \frac{1}{j\omega C_{iso,1}}, \tag{3}$$

$$Z_{Bearing,2} = \left( \frac{U_G}{U_{iso,\,2}} - 1 \right) \cdot \frac{1}{j\omega C_{iso,2}}, \tag{4}$$

where $Z_{Bearing,i}$ is the complex rolling bearing impedance, $U_G$ is the measured generator voltage, $U_{iso,\,i}$ is the measured voltage over the isolation, and $C_{iso,\,i}$ is the known capacitance of the isolation. The capacitances of the isolations are measured as $C_{iso,1} = 2.2$ nF and $C_{iso,2} = 2.6$ nF. The carrier signal frequency is set to 2.5 MHz, and the sampling rate is 50 MHz. The voltage amplitude is $\hat{U}_G = 2.5$ V [6].

The real part of the measured impedance signals is negative [5,6]. The authors explained this phenomenon as a calculation error because the isolations are assumed ideal. Modeling the isolations as not ideal turns the results into positive real parts, but they were not analyzed further to measure their real behavior [6], which leads to measurement uncertainties.

To avoid these uncertainties, another impedance measurement method has been applied to generate the validation test data. It is based on measurement bridges, using an alternating current as a carrier signal and gauged capacitors for the reference impedance. Figure 4 shows the equivalent circuit.

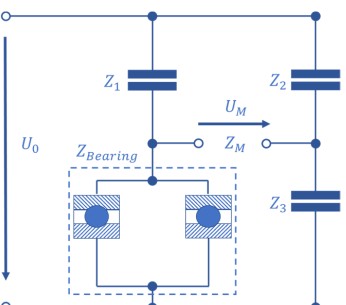

**Figure 4.** Equivalent circuit of the alternating current measurement bridge for impedance measurement.

The impedance of the bearings in the parallel connection $Z_{Bearing}$ is calculated using the following equation:

$$Z_{Bearing} = Z_1 \frac{Z_3 Z_M + [(Z_2 + Z_3)Z_M + Z_2 Z_3]\frac{U_M}{U_0}}{Z_2 Z_M - [(Z_2 + Z_3)(Z_M + Z_1) + 1]\frac{U_M}{U_0}}. \tag{5}$$

The reference impedance of the capacitors is tagged as $Z_1$, $Z_2$, and $Z_3$. The generator voltage is $U_0$ and the voltage at the oscilloscope is $U_M$. The resistance of the oscilloscope is $Z_M$. An open-short adjustment has been implemented to consider the influence of the measurement lead. To reduce parasitic effects, the carrier signal frequency is set to 25 kHz. The voltage amplitude is identical to the voltage divider. The sampling rate is set to 1 MHz.

This measurement approach has not been used before to detect rolling bearing impedance in fatigue tests. So, an important aspect is if the signals and features of the signals show the same behavior over the bearing operational time. This question will be addressed in this paper.

### 2.2. Test Rig and Impedance Measurement

All experiments are performed at the rolling bearing test rig of the Institute for product development and machine elements of the Technical University of Darmstadt. It contains four separate test chambers. In each chamber, four rolling bearings are located for observation. The test bench monitors the vibration, the temperature at every bearing, the motor torque, the revolution speed, and the lubricant temperature. Figure 5 shows one of the rig's test chambers. The test bench has an adjustable recirculating oil-lubrication system for each test chamber so that different lubrication conditions can be investigated.

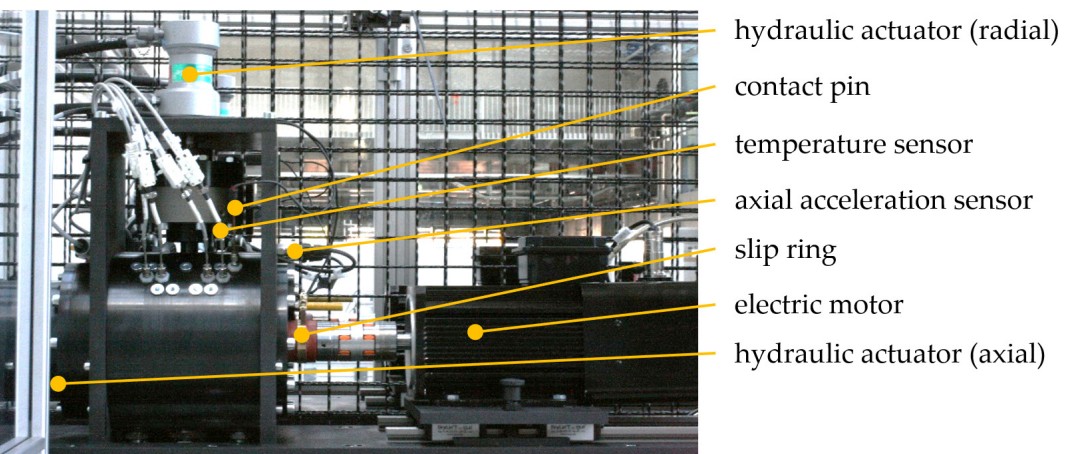

hydraulic actuator (radial)

contact pin

temperature sensor

axial acceleration sensor

slip ring

electric motor

hydraulic actuator (axial)

**Figure 5.** Test chamber of the bearing test rig.

The four bearings in a chamber are placed on the same shaft, which is electrically contacted using a slip ring. The bearing seats consist of two parts, separated by an insulating ceramic layer. A contact pin bypasses the insulation of the electrically observed test bearings. Within one of the chambers, two of the four bearings are investigated using impedance measurement methods. The exact configuration for the performed fatigue tests can be read in Martin et al. [5], which is also applied to the new measurement approach tested here for a better comparison. The bearings can be loaded with radial and axial forces by hydraulic actuators.

### 2.3. Design and Procedure of the Fatigue Tests

The individual feature selection procedure is applied to the data measured in five fatigue tests using the measurement method by Martin. The investigated bearings are angular contact ball bearings of the type 7205B-XL-TVP manufactured by FAG. These tests

were executed as part of earlier research at the Institute [6]. For validation purposes, another fatigue test is performed using the alternating current measurement bridge. Therefore, radial ball bearings of the type 6205-C-C3 from the manufacturer SKF are used and stressed under different conditions. A comparison of the bearing loads and test conditions of both measurement setups is displayed in Table 2. As mentioned before, a comparison of the impedance signals of both measurement methods is planned to obtain information about the quality of impedance signals for condition monitoring. All tests run under full lubrication, so the EHL contact and the capacitive electrical behavior in a normal operational stage can be ensured.

**Table 2.** Test conditions of the two varying measurement setups.

| Test Parameter | Investigation Tests [6] | Validation Test |
|---|---|---|
| Radial load | 3000 N | 7884 N |
| Axial load | 28,000 N | 3390 N |
| Dynamic safety | 0.95 | 1.92 |
| Speed | 4000 $\text{min}^{-1}$ | 5000 $\text{min}^{-1}$ |
| Oil temperature | 30 °C | 60 °C |
| Time between impedance measurements | 1 min | 2 min |
| Length of each impedance measurement | 1.34 s | 1.5 s |
| Carrier signal frequency | 2.5 MHz | 20 kHz |
| Carrier signal amplitude | 5 V peak to peak | 5 V peak to peak |
| Sampling rate | 50 MHz | 1 MHz |

*2.4. Preprocessing and Feature Generation*

First, the measured impedance data are preprocessed. Four time signals are directly calculated from the measured complex impedance signal, namely the real and imaginary parts and the absolute value and phase angle (see Table 1).

These four signals are further processed. Outliers are removed using a Hampel filter, whose mathematical explanation is described in [22]. A noise filter reduces noise due to the measurement setup and environmental influences. Using wavelets for noise filtering is especially effective when reducing noise while preserving abrupt changes with high-frequency components of the signal [23]. The impedance signal of a damaged rolling bearing is characterized by abruptly occurring peaks in the real and imaginary parts [5]. Therefore, preserving the high-frequency components of the signal is of high importance, which is why a wavelet filter is applied for noise reduction. To prevent misleading signal interpretation due to errors in the measurement setup, the mean value is removed. For this purpose, the mean value of the impedance signal in the run-in stage is subtracted from the signal itself.

In the following step, features are generated. They are derived from the time and frequency domain. For frequency domain features, it is necessary to calculate the frequency spectrum of the impedance signal by applying a discrete Fourier transform. For this purpose, a fast Fourier transform is used [3]. Compared to the time domain signal, the frequency spectrum often contains further information about the signal's properties [24]. In condition monitoring of rolling bearings, the frequency spectrum is particularly important for identifying the location and cause of the initial damage [6,24].

The process of feature generation is established in the field of condition monitoring of ball bearings using the vibration signal. Just like the impedance signal, the vibration signal of a damaged bearing is characterized by periodically occurring peaks during the rollover of the initial damage [1]. Due to this analogy, the state of the art of vibration signal feature engineering is applied to the impedance signal. The generated features listed in Table 3 are taken from studies dealing with vibration signals of rolling bearings [1,25,26]. The features of measurement are calculated for each of the four signals derived from the measured complex impedance signal. In total, this leads to 128 features for each impedance measurement.

**Table 3.** Features derived from the time and frequency domains.

| Number | Formula | Number | Formula |
|--------|---------|--------|---------|
| T1 | $T_m = \frac{\sum_{i=1}^{N} x(i)}{N}$ | F1 | $W_1 = W_{mf} = \frac{\sum_{k=1}^{K} s(k)}{K}$ |
| T2 | $T_{root} = \left( \frac{\sum_{i=1}^{N} \sqrt{|x(i)|}}{N} \right)^2$ | F2 | $W_2 = \frac{\sum_{k=1}^{K} (s(k)-W_1)^2}{K-1}$ |
| T3 | $T_{rms} = \sqrt{\frac{\sum_{i=1}^{N} (x(i))^2}{N}}$ | F3 | $W_3 = \frac{\sum_{k=1}^{K} (s(k)-W_1)^3}{K \cdot \left(\sqrt{W_2}\right)^3}$ |
| T4 | $T_{max} = max|x(i)|$ | F4 | $W_4 = \frac{\sum_{k=1}^{K} (s(k)-W_1)^4}{K \cdot W_2^2}$ |
| T5 | $T_{sd} = \sqrt{\frac{\sum_{i=1}^{N} (x(i)-T_m)^2}{N-1}}$ | F5 | $W_5 = W_{fc} = \frac{\sum_{k=1}^{K} (f(k) \cdot s(k))}{\sum_{k=1}^{K} s(k)}$ |
| T6 | $T_{skewness} = \frac{\sum_{i=1}^{N} (x(i)-T_m)^3}{(N-1) \cdot T_{sd}^3}$ | F6 | $W_6 = \sqrt{\frac{\sum_{k=1}^{K} (f(k)-W_5)^2 \cdot s(k)}{K}}$ |
| T7 | $T_{kurtosis} = \frac{\sum_{i=1}^{N} (x(i)-T_m)^4}{(N-1) \cdot T_{sd}^4}$ | F7 | $W_7 = W_{rmsf} = \sqrt{\frac{\sum_{k=1}^{K} \left(f(k)^2 \cdot s(k)\right)}{\sum_{k=1}^{K} s(k)}}$ |
| T8 | $T_{crest} = \frac{T_{max}}{T_{rms}}$ | F8 | $W_8 = \sqrt{\frac{\sum_{k=1}^{K} \left(f(k)^4 \cdot s(k)\right)}{\sum_{k=1}^{K} \left(f(k)^2 \cdot s(k)\right)}}$ |
| T9 | $T_{clearance} = \frac{T_{max}}{T_{root}}$ | F9 | $W_9 = \frac{\sum_{k=1}^{K} \left(f(k)^2 \cdot s(k)\right)}{\sqrt{\sum_{k=1}^{K} s(k) \cdot \sum_{k=1}^{K} \left(f(k)^4 \cdot s(k)\right)}}$ |
| T10 | $T_{shape} = \frac{T_{rms}}{\frac{1}{N} \cdot \sum_{i=1}^{N} |x(i)|}$ | F10 | $W_{10} = \frac{W_6}{W_5}$ |
| T11 | $T_{impulse} = \frac{T_{max}}{\frac{1}{N} \cdot \sum_{i=1}^{N} |x(i)|}$ | F11 | $W_{11} = \frac{\sum_{k=1}^{K} \left((f(k)-W_5)^3 \cdot s(k)\right)}{K \cdot W_6^3}$ |
| T12 | $T_{pp} = \max(x(i)) - \min(x(i))$ | F12 | $W_{12} = \frac{\sum_{k=1}^{K} \left((f(k)-W_5)^4 \cdot s(k)\right)}{K \cdot W_6^4}$ |
| T13 | $T_{var} = \frac{1}{N} \cdot \sum_{i=1}^{N} (x(i) - T_m)^2$ | F13 | $W_{13} = \frac{\sum_{k=1}^{K} \left(|f(k)-W_5|^{\frac{1}{2}} \cdot s(k)\right)}{K \cdot \sqrt{W_6}}$ |
| T14 | $T_{min} = \min(x(i))$ | F14 | $W_{14} = \sqrt{\frac{\sum_{k=1}^{K} \left((f(k)-W_5)^2 \cdot s(k)\right)}{\sum_{k=1}^{K} s(k)}}$ |
| T15 | $T_{wave} = \frac{\sqrt{\frac{1}{N} \cdot \sum_{i=1}^{N} |x(i)|^2}}{\frac{1}{N} \cdot \sum_{i=1}^{N} |x(i)|}$ | F15 | $W_{15} = max|s(k)|$ |
| T16 | $T_{peak} = \frac{T_{max}}{\sqrt{\frac{1}{N} \cdot \sum_{i=1}^{N} (x(i))^2}}$ | | |
| T17 | $T_{LI} =$ $\sum_{i=1}^{N} \sqrt{(x(t_i + \Delta t_s) - x(t_i))^2 + \Delta t_s^2}$ $\approx \sum_{i=1}^{N} |x(t_i + \Delta t_s) - x(t_i)|$ with sampling period $\Delta t_s$ | | |

The time series with $i = 1, 2, 3, \ldots, N$ is $x(i)$ while $s(k)$ is a frequency spectrum with $k = 1, 2, 3, \ldots, K$. $K$ is the total number of spectral lines in the spectrum, and $f(k)$ is the frequency value of the $k$-th spectral line.

Figure 6 summarizes the elaborated steps to derive features from the measured complex impedance signal.

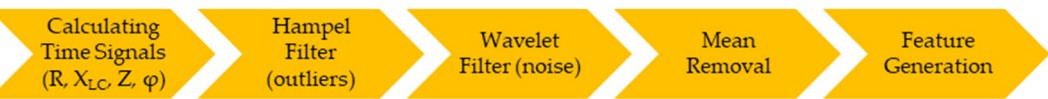

**Figure 6.** Preprocessing and feature generation process.

*2.5. Individual-Feature Selection*

The suitability of these generated features for use in condition monitoring is assessed. Therefore, individual-feature selection is applied. This method ranks the features based on a specific criterion [12]. In this case, the criterion is supposed to quantify the ability of a feature to draw conclusions about the condition of the observed rolling bearing.

The conditions of the dynamically stressed bearing are calculated by assuming the hypothesis of linear damage accumulation. Accordingly, the total damage is calculated by summing up the damage portions $q_i$ of each cycle [27]. For the calculation of these damage portions, the duration of the load level $t_i$ and the basic rating life $L_{10h}$ are divided as shown in formula 3.1 [28].

$$q_i = \frac{h_i}{N_{SSZ,i}} = \frac{t_i}{L_{10h} \cdot 60\frac{min}{h} \cdot 60\frac{s}{min}}.$$ (6)

The basic rating life is calculated by the speed $n_{rpm}$ of the bearing, its dynamic load capacity $C$, and life exponent $p$, as well as the dynamic equivalent load $P$ (see formula 3.2) [29,30]. The dynamic equivalent load depends on the rolling bearing geometry and the radial and axial loads [29]. The test rig records the loads and speed of the bearing during the fatigue tests.

$$L_{10h} = \frac{10^6}{60\frac{min}{h} \cdot n_{rpm}} \cdot \left(\frac{C}{P}\right)^p.$$ (7)

The total damage of the bearing can be calculated for the time of each impedance measurement using the recorded operational parameters. The time of initial damage detection of the five fatigue tests scatters a lot. As a result, the total damage of the bearings at the end of the tests differs widely. To obtain a universal measure for the bearing condition, a min-max scaling algorithm normalizes the total accumulated damage (see formula 3.3 [31]). This leads to the normalized accumulated damage, which rises from zero to one during a fatigue test.

$$D^*(m) = \frac{D(m) - D_{min}}{D_{max} - D_{min}}.$$ (8)

The criterion for the individual-feature-selection process expresses the relationship between a considered feature and the normalized accumulated damage. This relationship can be quantified by their correlation coefficient [32]. The correlation coefficient, according to Bravais–Pearson, is used to find the strength of the linear relationship between two variables [33]. To consider monotonic, nonlinear relationships, the correlation coefficient, according to Spearman, is used [33]. Features with high correlation coefficients with normalized accumulated damage are considered probable indicators of bearing damage. After calculating the correlation coefficients, each feature is ranked according to its Bravais–Pearson and Spearman correlation coefficient. The final ranking of features is achieved by considering the average rank of a feature resulting from the two mentioned criteria.

The individual feature selection is performed twice at different time intervals. Firstly, all measured data are taken into consideration; thus, the whole lifespan of the tested bearing is observed. Secondly, the last hour before initial damage detection by the test rig is exclusively studied. This enables the observation of the behavior of a feature before initial damage without the influence of the pre-run-in stage. This is of high interest because this stage shows similar effects in the impedance signal to the impedance signal after the initial damage [6].

## 3. Results

The resulting correlation coefficients of each feature with the label, sorted by their rank in the respective ranking, are displayed in Figure 7. A group of features characterizes the resulting distribution with high correlation coefficients compared to the remaining features.

Since the ten highest-ranked features stand out with a particularly high correlation in both rankings, these features are considered for further investigation.

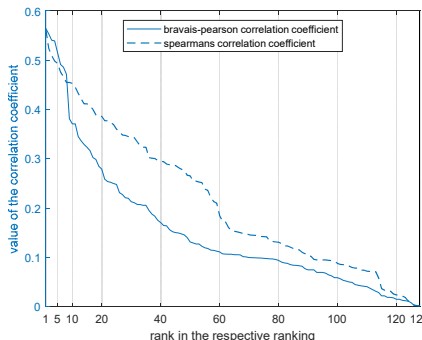

**Figure 7.** Correlation coefficients of each feature with the label.

The features that appear in both of the chosen subsets are ranked according to their average rank, as described in the previous chapter. The resulting three highest-ranked features are shown in Table 4. Since the procedure has been applied at two different time intervals, two rankings are depicted.

**Table 4.** Ranking of individual features.

| Rank | Whole Lifespan | Last Hour |
|------|----------------|-----------|
| 1. | Feature 88: RMS frequency (F7) of the absolute value | Feature 102: skewness (T6) of the phase angle |
| 2. | Feature 56: RMS frequency (F7) of the imaginary part | Feature 60: skewness of the frequencies (F11) of the imaginary part |
| 3. | Feature 86: central frequency (F5) of the absolute value | Feature 92: skewness of the frequencies (F11) of the absolute value |

### 3.1. Description of Individual Features

In the following chapter, the top-ranked features listed in Table 4 are plotted and described in detail. The observations are further explored in Section 4.

First, the features considering the whole lifespan of the tested bearings are examined. The three highest-ranked features are correlated with each other. Their Bravais–Pearson correlation coefficients are greater than $r = 0.98$, with a deviation of approximately $\pm 0.009$ according to the 95% confidence interval. Thus, only the feature on rank one, namely the root-mean-square (RMS) frequency of the absolute value of the impedance, is representatively examined. The normalized value of this feature over the label is displayed in Figure 8. The feature measurement series of the five fatigue tests using the measurement setup by Martin are depicted in different colors. Three intervals are shown to allow a closer view of the features in the first and last hour of the fatigue test.

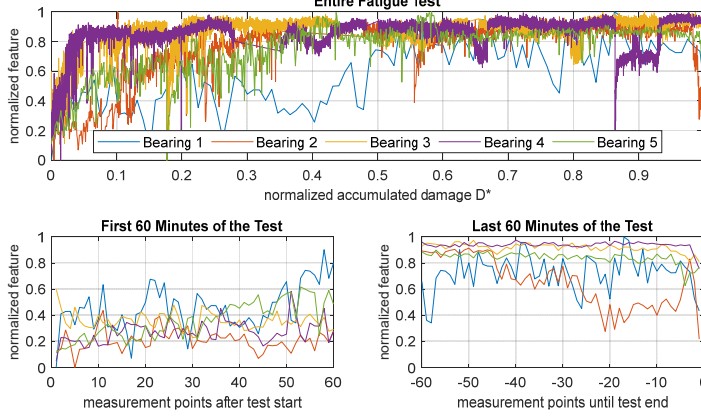

**Figure 8.** RMS frequency of the absolute value of the impedance.

The behavior of this feature is described in the following paragraph. Starting at a low level in the pro-run-in stage, the feature value increases until shortly before the end of the test. In all fatigue tests, a strong decline of the feature from the last 2 to 45 min before the end of the test is observed. The described three phases across the bearing lifespan are not clearly separated but rather connected to each other by a transition of the feature value. This behavior can be seen most prominently at bearing two.

In Figure 8, some noticeable abnormalities and characteristics are addressed in the following and explained in Section 4. At the beginning of the test, the feature values of the different tests seem to rise at a different pace. Also, there are visible gaps in the graphs, like at bearing 2 in the range of the normalized damage from 0.35 to 0.48 and at bearing 4 from 0.28 to 0.35. Another abnormality can be seen at bearing 3 at approximately $D^* \approx 0.18$ and at bearing 4 at $D^* \approx 0.86$. There, the features suddenly jump to a new level on which they remain for a while.

Now the features, seen as a probable indicator of bearing damage, considering the last hour before the end of the fatigue test, are described. First, the skewness of the phase angle of the impedance is shown in Figure 9.

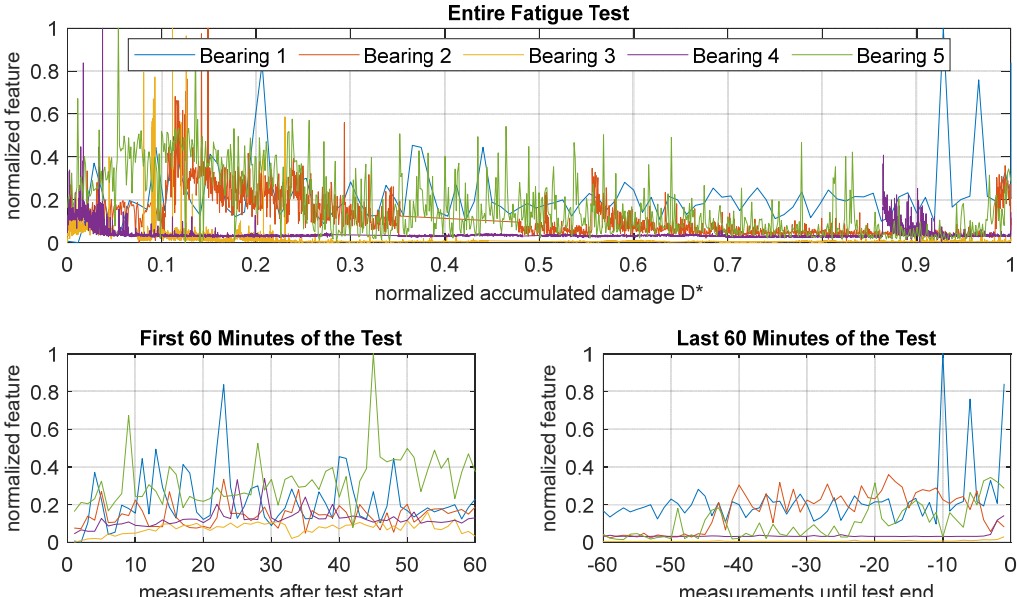

**Figure 9.** Skewness of the phase angle of the impedance.

At the beginning of the tests, high peaks and noisy behavior are observed. Next, the feature declines to a lower level and significantly less noise. At the fatigue test end, the feature abruptly rises significantly. Again, this feature description indicates three phases of different feature behavior with gradual transitions in between.

The features bearing 2 and 3, considering the last hour of the test, show the same behavior, which is confirmed by the Bravais–Pearson correlation coefficient of $r = 0.99$ with a deviation of approximately $\pm 0.014$ according to the 95% confidence interval. So, only describing the feature on rank two, shown in Figure 10, is sufficient.

The skewness of the frequency values is weighed by the amplitudes of the corresponding frequencies, as shown in Table 3. In the beginning, the fatigue tests show a different behavior. Thus, one can see a significant rise in the feature at the beginning to different levels for all test results. For the rest of the fatigue test, the feature declines steadily. A roughly linear behavior can be observed. The absolute values vary significantly.

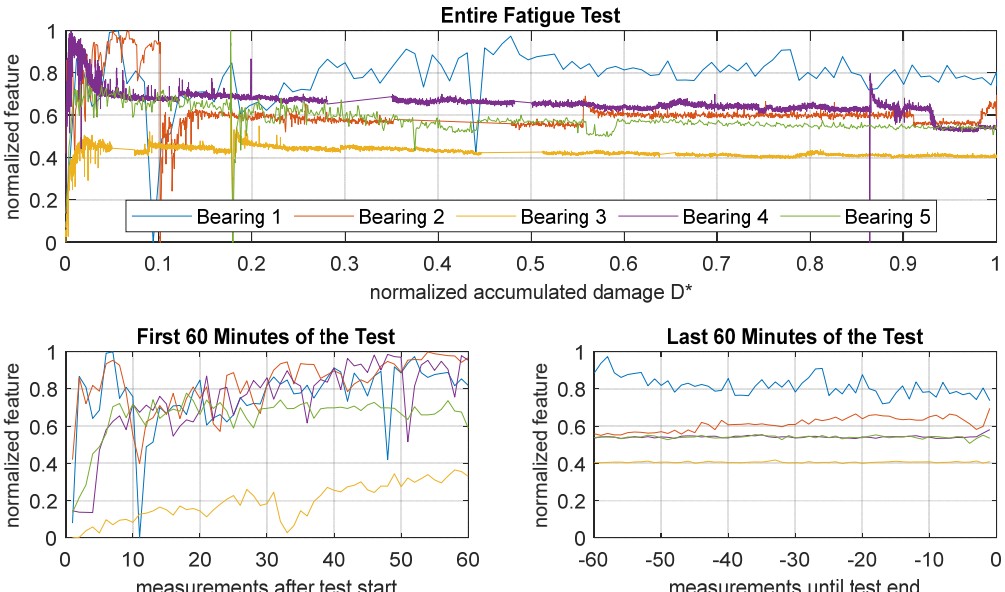

**Figure 10.** Skewness of the frequency values imaginary part of the impedance.

### 3.2. Validation Fatigue Tests

The results are validated by analyzing the validation fatigue test with the alternating current measurement bridge and different test setup parameters, as described in Section 2.3. Special attention should be drawn to the deviating time between measurements, which is two minutes. At this fatigue test, none of the test bearings but one of the support bearings failed. This support bearing is insulated electrically by ceramic rolling elements, thus not directly influencing the measured impedance. After initial damage detection by the test rig, the test was continued for another 60 min to obtain a higher number of measurements of the damaged bearing. The bearings were not disassembled during the entire test to exclude any fault effect. The features from Table 4 are now shown for the validation test.

In the first considered interval, the whole lifespan, the first two features again have a very high correlation of 0.9995, so, just the highest ranked feature is shown in the following graph in Figure 11.

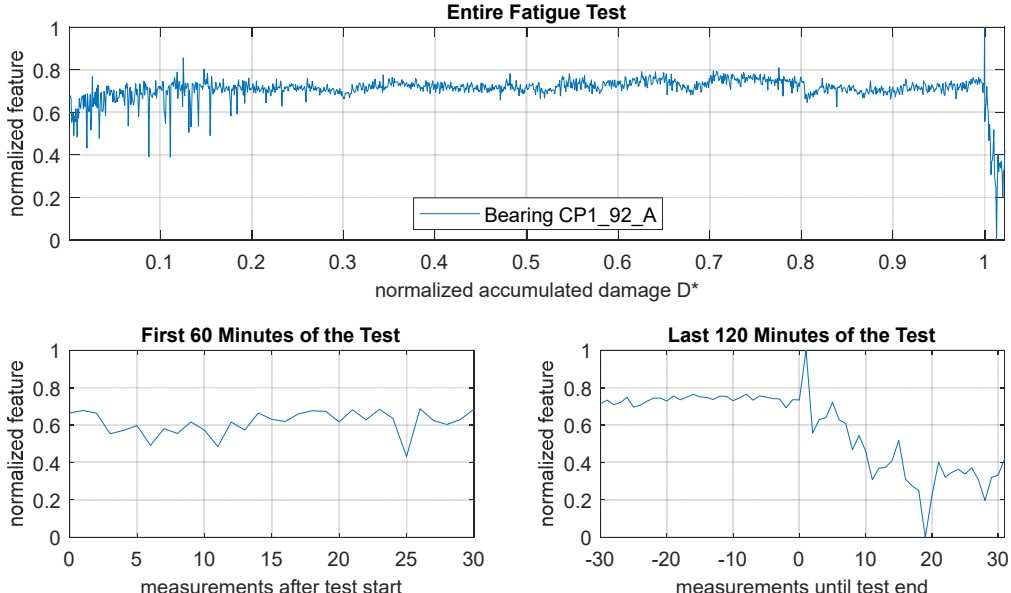

**Figure 11.** RMS frequency of the absolute value of the impedance (validation test).

The feature behavior corresponds to the behavior previously observed at the fatigue tests by Martin. It starts at a low level, inclines steadily, and suddenly declines at the initial pitting damage. In particular, the decline is clearly visible and very strong. It is particularly interesting that the feature only declines after initial damage detection, whereas in Figure 8, the feature showed abnormalities before the initial damage detection by the test rig. With the continuing load in the validation test, even after initial damage detection, the feature drops way below the level of the pre-run-in stage.

In contrast to the previous features, rank three only possesses a correlation coefficient of $r \approx 0.65$ with ranks one and two, making it necessary to examine this rank additionally. The central frequency of the absolute value of the impedance is depicted in Figure 12. Looking closer at this feature, the difference in the feature's behavior using the measurement setup by Martin becomes obvious: The feature behaves differently in the pre-run-in stage. At the test beginning, the feature is located on the highest level, declines to a lower level in standard operation, and finally drops after initial damage. With this behavior, the feature possibly enables the distinction of run-in and damaged phase using the measurement bridge method. This makes the feature possibly feasible for a distinction between the three bearing life phases that could be observed in the previous chapter.

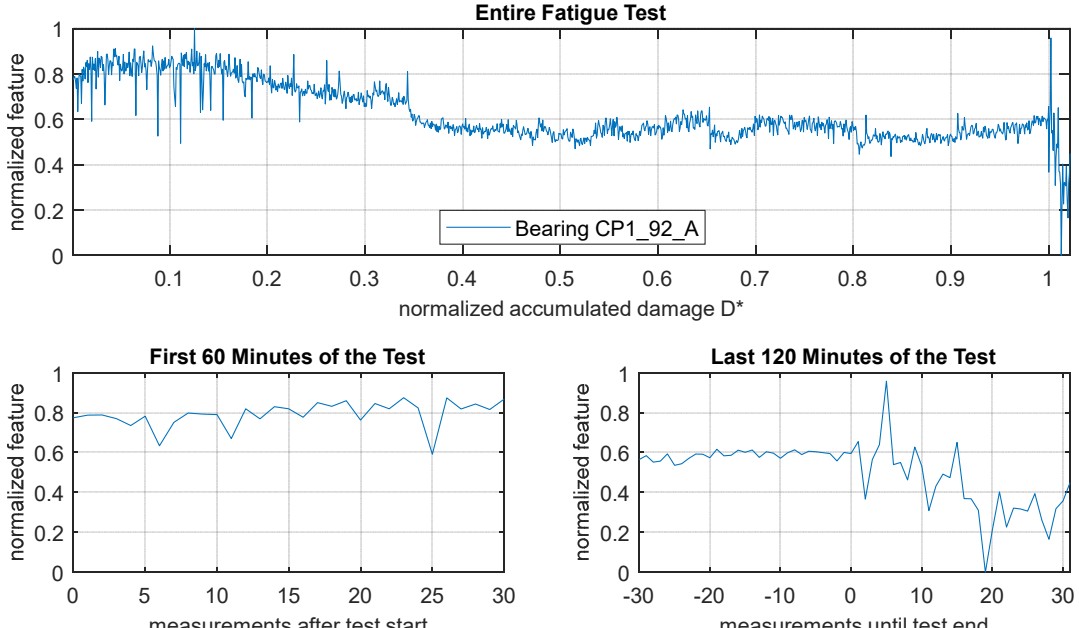

**Figure 12.** Central frequency of the absolute value of the impedance (validation test).

Looking at the highest-ranked features considering the last hour of the fatigue test, the same phenomena as in the already described measurement series are observed. Nevertheless, the significant indications of bearing damage are again only visible after the initial damage detection and not in advance. The feature on rank one is shown in Figure 13 and confirms the expected feature behavior. Just as in the previous chapter, three phases are clearly visible in the feature behavior.

The features of bearings 2 and 3 again show a very high correlation, and rank two can be seen in Figure 14. The feature again rises at the beginning of the test and declines with further damage progression. However, the decline is much noisier than in Figure 10 and does not show a linear behavior. A significant decrease in the feature can be observed after initial damage detection.

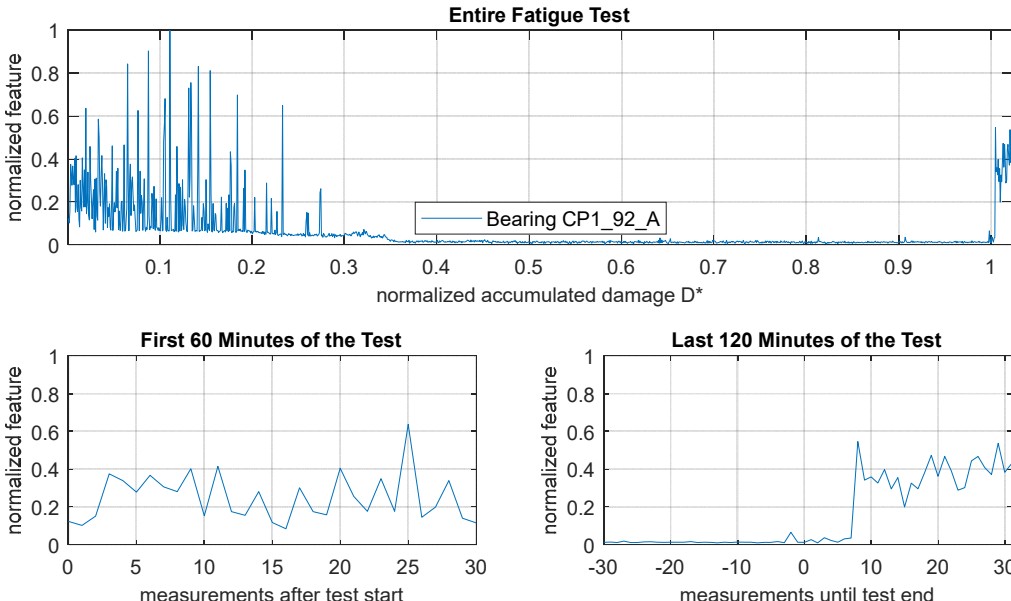

**Figure 13.** Skewness of the phase angle of the impedance (validation test).

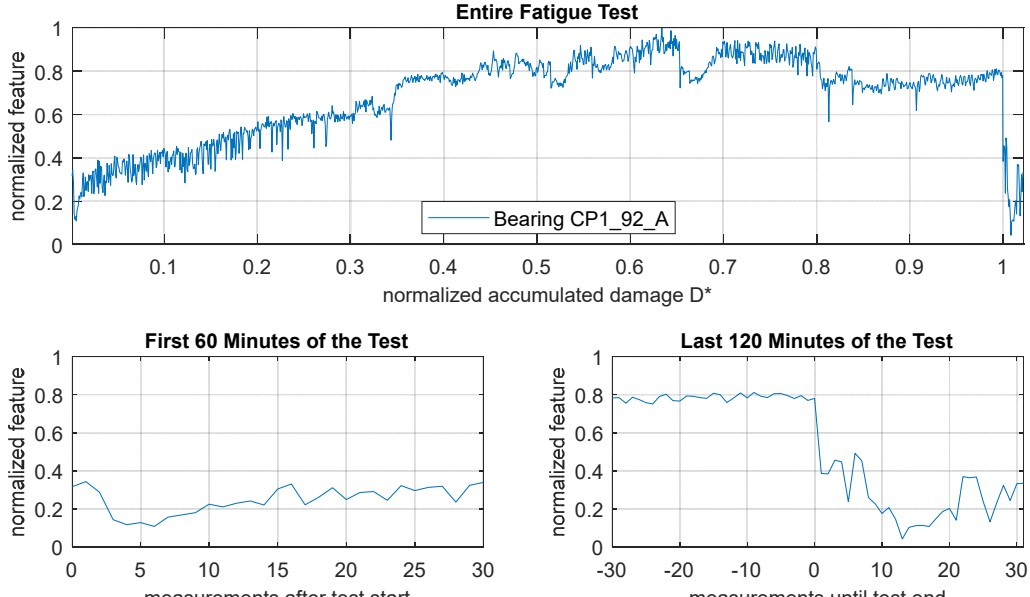

**Figure 14.** Skewness of the frequency values imaginary part of the impedance (validation test).

### 3.3. Comparison to Vibration Signals

The vibration signals are automatically recorded by the test rig, and the introduced features of the impedance signals are compared in terms of their suitability as an indicator of rolling bearing damage. The vibration data of bearing 5 show a significant increase in amplitude three minutes prior to the end of the fatigue test. The other tests show a significant increase only 30 s before the test ends (see Figure 15).

In contrast, the most significant features (rank one in each time interval) show a conspicuous behavior way before the vibration signal. As illustrated in Figure 15, initial damage can be recognized by a decrease in the RMS frequency of the absolute value of the impedance. Most bearings show that effect in the last two minutes before the test ends. Bearing 5 shows a significant decrease three minutes before the test ends, and bearing 2, even 40 min before the initial damage detection and test are stopped by the test rig.

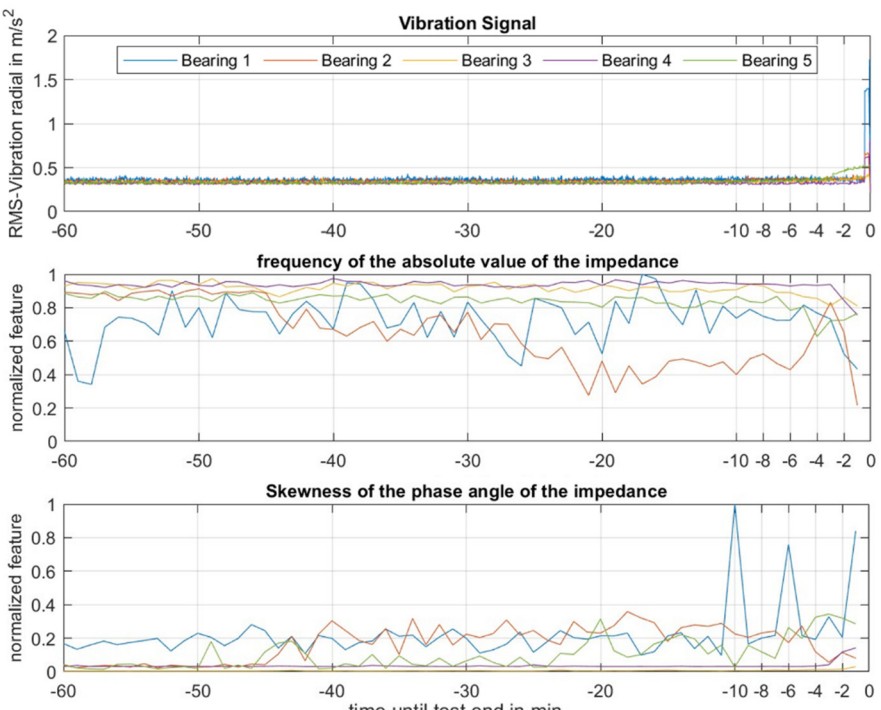

**Figure 15.** Vibration signals and impedance features in the last 60 min of the fatigue tests.

Likewise, the skewness of the phase angle enables early damage detection. This feature shows an increase in amplitude before the breakdown. Again, the noticeable behavior is seen at bearing 2 40 min before the test ends. Also, bearing 1 shows an increase in feature amplitude nine minutes before the end and bearing 5 more than 45 min before the end of the experiment. The difference in the behavior of the two features regarding damage detection may be affected by different causes, types, or progression of the detected bearing damage.

All in all, the features enable a detection of the bearing damage prior to the vibration signal. Especially the combination of multiple features could be useful to exploit the full potential of the different features for the detection of certain types of bearing damage. Nevertheless, further development of the signal measurement setup and signal processing should be considered to improve the accuracy of damage detection.

## 4. Discussion

In this section, the results of Section 3 will be interpreted and discussed. The validation is carried out on independent data sets not included in the previously gathered data. Afterward, the results of the validation test will be compared to the results of the investigation tests. In the end, the time gap between the vibration data and impedance features will be explained.

### 4.1. Phenomenological Explanation

The behavior of the individual features observed in the sections before is interpreted in the following. Along with it, phenomenological explanations of the feature's significance and characteristics are given while taking the available knowledge of the electrical properties of lubricated rolling bearings into account. The explanatory approaches in this chapter obtain a relation between electrical and tribological phenomena to the observed feature behaviors.

Table 4 shows different high-ranked features when looking at the last hour of the fatigue test compared to the whole bearing lifespan. In conclusion, some features are especially significant in the time interval shortly before initial damage, while others possess more information about the damage progression over the whole bearing life. This explanation is plausible since the pre-run-in stage shows similar behavior to the signal before

initial damage [6]. Consequently, features that are significant in the last hour of the test may not be significant over the whole lifespan due to the influence of the pre-run-in stage.

From the individual features, the RMS frequency in the spectrum of the absolute value of the measured impedance is interpreted first. According to the chosen criterion, it is the most significant feature regarding overall extracted data during the whole fatigue test. It correlates strongly with the RMS frequency in the imaginary part and the central frequency of the absolute impedance value. Thus, there seem to be phenomena underlying these features that cause their highly similar behavior. This assumption is enforced by the observation of previous investigations that show similar effects in the real and imaginary parts of the bearing impedance when it comes to damage progression [5]. In the following, an approach to explain the characteristic behavior of that feature is introduced.

In the pre-run-in stage, the roughness of the bearings' running surfaces is high; the contact of roughness peaks results in high noise of the impedance signal. The noise often appears with amplitudes of similar magnitude [5]. These seemingly periodically occurring effects lead to low frequencies in the frequency spectrum of the impedance measurement (rough dimension of 1–10 kHz). In the frequency spectrum of impedance measurements in the pre-run-in stage, there are high amplitudes of these frequencies observable (see frequency spectrum at $D^* = 0$ in Figure 16). The same effect is seen in the spectrums of the other tested bearings, too.

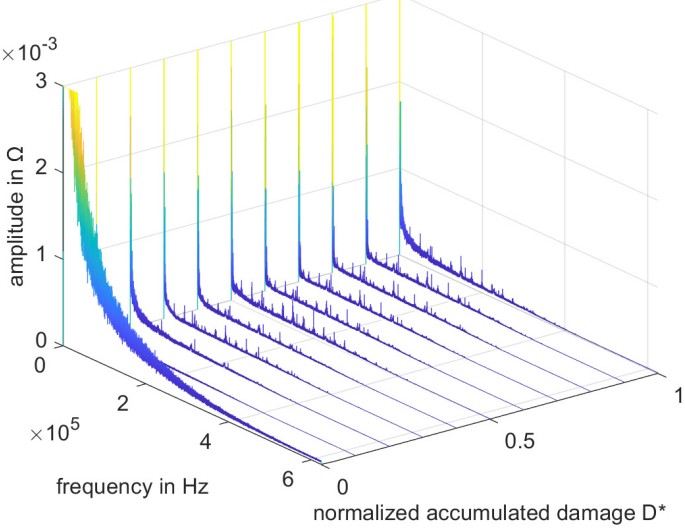

**Figure 16.** Waterfall diagram of the absolute impedance value of bearing 5.

In the run-in stage, the surface roughness peaks are smoothened, resulting in fewer electrical breakdowns and less noise in the impedance signal. The low frequencies, induced by the high surface roughness, are no longer present, which leads to a higher central and RMS frequency according to the corresponding formula (see Table 3). In Figure 16, this is visible in the decline of amplitudes of the low frequencies with progressing total damage.

The bearing failure due to pitting starts with a crack underneath the contact surface [34]. During the rollover of this beginning bearing damage, impulses in the real and imaginary parts of the impedance signal can occur, similar to those in the pre-run-in stage [5]. These can be caused by the temporary breakdown of the isolating lubricant film during damage rollover. These impulses could again lead to low frequencies in the spectrum. This effect can be observed in Figure 16 in the spectrum right before initial damage detection at $D^* \approx 1$. This again causes a decline in the RMS frequency and the other investigated features, explaining the described behavior before the initial damage detection by the test rig.

Now, after interpreting the rough trend of these features, the described abnormalities in the graphs are addressed as well. The seemingly different paces at which the features rise in the different fatigue tests originate in an approximately constant run-in duration

and differing fatigue test times. As a result, the duration of the pre-run-in stage relative to the fatigue test time correlates to the fatigue test time itself, leading to the described effect. Visible gaps in the graphs are caused by a temporary failure of the impedance measurement system. For the duration of the gap, no impedance measurements exist, while the bearing has been damaged continuously, which is captured by the test rig and considered with the calculation of the normalized total damage. Sudden jumps of the feature are caused by a stop of the test rig, including the disassembly of the test rig. These disassembly processes are necessary in order to exchange the failed test bearing and unavoidably lead to inaccuracies in the impedance measurements of the second test bearing. To avoid this effect in future experiments, disassembly during tests should be avoided. This can be achieved by exchanging both test bearings after a detected bearing damage instead of exchanging only the damaged one.

Next, the most significant feature in the last hour before initial damage detection is explained, which is named the skewness of the phase angle of the impedance signal in the time domain. The high skewness at the beginning can be seen as an indicator of the pre-run-in stage since the duration of the high skewness shows the same effects as the duration of the pre-run-in stage described in the previous paragraph.

Before the end of the fatigue test, the skewness again shows distinctly higher values. The high feature values in both the pre-run-in and damaged phase can be phenomenologically explained: A positively skewed distribution describes a graph as shown in Figure 17 [33].

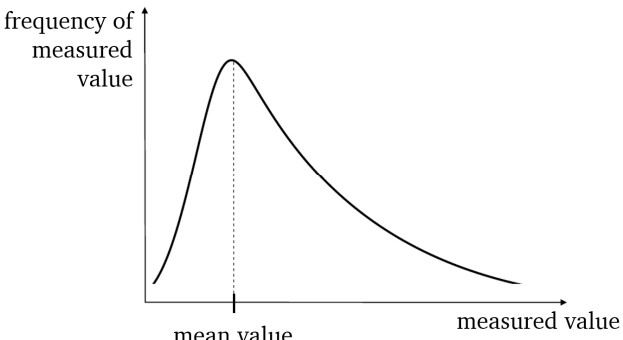

**Figure 17.** Distribution with positive skewness [35].

In the impedance measurement, the steep rise at the left side of the distribution is caused by the clustering of measurement points with an impedance characteristic for an elastohydrodynamic (EHL) contact. This form of contact is the case in the run-in stage with no electrical breakdowns. Because of the capacitive characteristic of a lubricated bearing with EHL contact [15], the phase angle is approximately $\varphi \approx -90°$, as seen on the left side in Figure 18.

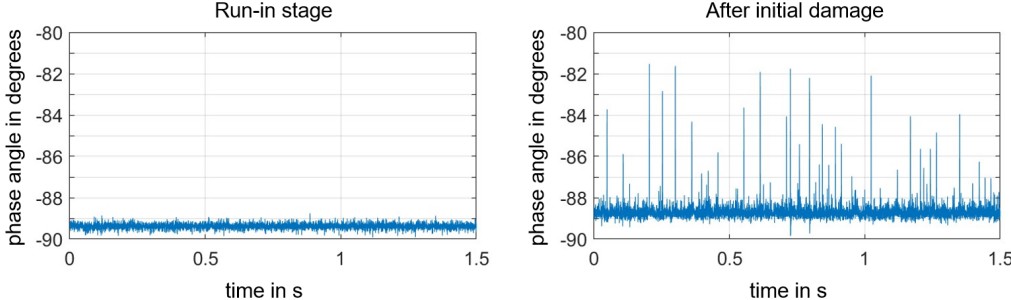

**Figure 18.** Phase angle of individual impedance measurements of the validation test.

Electrical breakdowns in the pre-run-in and damaged phases lead to signal changes of the impedance with resistive characteristics (see Figure 18 right). Resistive behavior

leads to a phase angle closer to zero, which, in this case, results in positive impulses of the phase angle. These positive impulses can be seen in Figure 17 in the form of measurement amplitudes on the right-hand side of the mean value; the more resistive the behavior, the higher the skewness of the phase angle.

The other features in the ranking in Table 4 do not show behavior that clearly shows a distinction between the different test stages. In conclusion, no explanation approach for this behavior is presented in this publication. Nevertheless, the approximately linear behavior of the features in the run-in stage can be useful in damage-progression detection, although the absolute value does not offer as much information about the damage.

In conclusion, the most significant features resulting from the individual feature selection are features in the frequency domain. The features from Table 4 seem to have a relationship with the chosen label. They possess the potential to be useful for early damage detection at rolling bearings and show explainable connections between the bearing impedance and the bearing damage state. It can be said that the information extracted from the impedance signal could be enlarged compared to Martin et al. using this simple feature engineering approach. Therefore, it could be possible to use classification algorithms for more accurate differentiation of different bearing health conditions.

Other impedance-based monitoring approaches focus on the lubrication condition in the EHL contact, e.g., Barz and Maruyama et al. [7,14]. Different methods are used to investigate the lubrication film thickness in rolling element bearings, but they do not include the bearing health condition over its lifespan; this differentiates the approach presented in this paper from the other impedance measurement methods.

### 4.2. Effects Observed in the Validation Test

In contrast to the findings observed in the other fatigue test, the validation test only showed the expected feature behavior during or after peaks occurring in the vibration signal. Nevertheless, this does not disprove the validity of the impedance features for early rolling bearing detection. This effect is caused by the circumstance of a support bearing failing instead of a test bearing. Thus, none of the bearings, whose impedance has been measured directly, failed, and the impedance consequently remained stable at first.

The support bearing failure may have led to higher stress at the test bearing because of vibrations, impacts, and possible load redistribution. The higher stress of the lubrication film in the test bearings may result in an affinity to metallic contact and, consequently, to resistive behavior. This would explain the phenomena observed in Section 3.2 despite the support-bearing failing. In addition, since the test-bearing impedance is influenced only by a support-bearing failure because of vibrations, the comparatively late impedance feature response in the validation fatigue test can be explained, too.

For both measurement approaches, the same feature was selected. The behavior of the features in both cases was identical with one exception. That means the impedance signal and its features are independent of the measurement approach used to record them. Even if the validation test did not detect damage at a test bearing, failures in other components can be seen in the signal. So, the impedance measurement might be used for condition monitoring not only for rolling bearings but also for other machine elements interacting with the shaft the observed bearings are located at.

Another aspect is that the same features are selected for different rolling bearing types in both tests. The original test was executed using angular groove ball bearings of type 7205. The validation test used deep groove ball bearings of type 6205. Because the feature in both cases showed the same behavior with one exception, it can be said that the impedance is bearing type independent. To explain the signal and feature behavior more precisely, further investigation is needed with a higher variance of bearing types.

### 4.3. Explanation of Delay between Vibration and Impedance Features

In this section, the possible causes of the delay between the rise of the vibration signal and the observed phenomena in the impedance feature signals shall be examined.

The higher vibrations of a damaged bearing are caused by a crack in the runway surface. During the bearing balls roll over the crack, vibrations are created [1]. However, the bearing impedance might be more sensitive to bearing damage in the early stages. When the crack forms underneath the runway surface, this beginning bearing damage might already have an impact on the electric transition behavior, which would cause changes in the measured impedance signal.

In further research, a comparison of the impedance signal features to the advanced vibration analysis and motor current analysis is necessary. Other papers could show the possibilities of vibration analysis using, e.g., deep feature learning [20]. They are material independent, as mentioned in Section 1.3, which allows a broader application field. In the case of motor current-based condition monitoring, additional sensors are not needed, which is an important cost factor. Impedance-based condition monitoring is applicable for slow-rotating machinery or critical processes and systems [36]. For a higher data quality, disturbance factors have to be identified and analyzed. For system applications, the exact electrical paths through the structure have to be known. First, the results show that disturbance factors have a specific behavior [37] that requires further investigation to be applicable in real applications. A remaining useful life (RUL) prediction for rolling element bearings is not investigated yet. Based on the results discussed before, there is a possibility that the impedance features can be used for RUL prediction. To research this topic, additional fatigue tests are necessary, as well as additional feature engineering methods.

## 5. Conclusions

The aim of this work was the investigation of impedance signals and their features over the operational time of rolling bearings. The impedance signals have been preprocessed, and individual feature selection was used to extract a higher amount of information from the signals. The features have been analyzed in the time and frequency domain based on the state of research for vibration data. Three phases could be identified in the operative life of a bearing, according to early research. Phenomenological explanations of the feature behavior were derived. In all five fatigue tests, the impedance signal changed before the vibration signals of the test rig sensors showed abnormalities. To clarify this, further research is necessary with a higher amount of fatigue test data. In addition, impedance features in the time-frequency domain have not been investigated yet.

Because uncertainties in the five impedance signals occurred, a more robust measurement approach has been developed and tested in an additional fatigue test. The selected features of both measurement approaches showed the same behavior over the bearing operational life. So, there is the possibility that impedance features map the bearing life independently from the measurement principle. In the validation test, the test bearings did not fail, but the support bearings did. The impedance features changed analog to the vibration signals, which means that the impedance measurement is able to detect damages not only at the observed bearings but also at machine elements located on the same shaft.

In the different test setups, two different bearing types were investigated. Because the impedance shows nearly the same behavior over the bearing's lifetime with one exception, it is possible that the impedance-features behavior is bearing-type independent. Further research is necessary to investigate this phenomenon and explain the feature's behavior.

In summary, it could be shown that impedance measurement can be used for condition monitoring of technical systems. Further research is needed to deepen the understanding of rolling bearing impedance and the features calculated from it. It is also possible to use machine learning algorithms for further investigation. Therefore, more fatigue tests with different operational parameters are necessary. In addition, the changes in bearing type and scale have to be investigated to ensure that the impedance is independent of these factors. In this paper, ball bearings have been used as test bearings. In the future, roller bearings and bearings with line contact, in general, must be examined.

The results presented in this paper show the opportunities for impedance-based condition monitoring. As mentioned in Section 4.3, the technique can be applied for special-use cases where vibration analysis is not sufficient for condition monitoring. The implementation of industrial gearboxes for their observation is already addressed at the Institute. Indicators could be found that the impedance measurement is able to observe not only the rolling element bearings themselves but also the entire gearbox. In this case, further research about the impedance behavior is needed.

**Author Contributions:** Conceptualization, F.M.B.-D., Q.S.K. and E.K.; methodology, F.M.B.-D. and Q.S.K.; software, F.M.B.-D. and Q.S.K.; validation, F.M.B.-D., Q.S.K. and E.K.; formal analysis, E.K.; investigation, F.M.B.-D. and Q.S.K.; resources, E.K.; data curation, Q.S.K.; writing—original draft preparation, F.M.B.-D. and Q.S.K.; writing—review and editing, E.K.; visualization, Q.S.K.; supervision, E.K.; project administration, F.M.B.-D. and E.K.; funding acquisition, E.K. All authors have read and agreed to the published version of the manuscript.

**Funding:** The authors thank the Deutsche Forschungsgemeinschaft (DFG, German Research Foundation), which funded the presented research within the project "Early damage detection of rolling bearings by electric impedance measurement". Funded by the Deutsche Forschungsgemeinschaft (DFG, German Research Foundation)—463357020, Gefördert durch die Deutsch Forschungsgemeinschaft (DFG)—463357020.

**Data Availability Statement:** The authors can be asked for the data used in this paper.

**Conflicts of Interest:** The authors declare no conflict of interest. The funders had no role in the design of the study; in the collection, analyses, or interpretation of data; in the writing of the manuscript; or in the decision to publish the results.

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
