# Peer review of "Individual Feature Selection of Rolling Bearing Impedance Signals for Early Failure Detection"

_lubricants, doi:10.3390/lubricants11070304_

Round 1

Reviewer 1 Report

Review report attached

Need proof reading

Author Response

We authors thank you for your revision. With your feedback, we were able to improve our paper. Written below you'll find our answers to your comments.

The writing is like a research report instead of a paper. For example, the introduction of the Feature Engineering section should be concise, and the Conclusion only describes the key findings. Numerous mentions of the affiliation, such as “The impedance data have already been generated in early research at the institute for product development and machine elements of 589 the University of Technology Darmstadt,” is unnecessary.

We checked the language and improved it.

The authors mentioned the machine learning approach in the individual feature selection, but it needs to be clarified if or how it is applied.

Machine learning was an example. It was to confusing, so the part was deleted.

The lubrication condition of the EHL contact is essential for the electric behavior of the bearing. Therefore, it needs more background and discussion on these (e.g., doi:10.1115/1.4007809, doi:10.1115/1.2920859).

The lubrication conditions and their influence on the electric behavior is presented in section 1.3. It has been extended considering the phase angle behavior to clarify the behavior of the phase angle skewness in section 4: “In case of capacitive behavior, the phase angle tends to -90°, which can be used as an indicator for lubrication condition. For phase angles about 0°, an ohmic behavior can be observed and metallic contacts occur [14, 18, 19].” It has also clarified that all test were run under fully lubrication film thickness: “All test were run under full lubrication, so the EHL contact and the capacitive electrical behavior in normal operational stage can be ensured.”

The remaining useful life is fundamental to condition monitoring. So, an elaboration on this is critical to enhancing the significance of the manuscript (e.g., doi:10.1007/s00170-022-09377-9, doi:10.3390/s16060795).

A RUL prediction is not part of the research presented in this manuscript. Three different condition states could be identified looking at the impedance features, which is presented in the results. This has been clarified in the manuscript at the end of section 4: “A remaining useful life (RUL) prediction for rolling element bearings is not investigated yet. Based on the results discussed before, there is the possibility that the impedance features can be used for RUL prediction. To research this topic, additional fatigue tests are necessary.“

The authors should discuss this study’s significance compared to previous studies using impedance signals.

It can be said, that the information density of the impedance signal could be enlarged compared to MARTIN ET AL. using this simple feature engineering approaches. So it could be possible to use classification algorithms for a more accurate differentiation of different bearing health conditions.

Other impedance based monitoring approach are focusing on the lubrication con-dition in the EHL contact, e.g. BARZ and MARUYAMA ET AL. [7, Quelle]. Different methods are used to investigate the lubrication film thickness in rolling element bear-ings, but they do not include the bearing health condition over its lifespan. That dif-ferentiates the approach presented in this paper from the other impedance measure-ment methods.

The lubrication system is a crucial element of the rolling bearing. As such, it should be discussed (e.g., doi:10.1243/1350650011541783, doi:10.3390/lubricants11030136).

The importance of the lubrication systems for rolling element bearings is crucial, indeed. For the approach presented in this study, it is connected to the lubrication system of the rolling bearing test rig, which is already mentioned under feedback topic 3.

Reviewer 2 Report

This work is focused on the field of condition monitoring applied to the detection and identification of faults in bearing elements. Specifically, the impedance signal of five fatigue tests has been investigated using individual feature selection.

The proposal is interesting; however, some issues must be addressed.

1. In literature review presented in Section 1 presents some of the most relevant research that have been already published, indeed, the proper organization of Section 1 allows to understand challenges that are mandatory to overcome. In this sense, new material technologies have emerged, and it could be interesting to include a brief paragraph to mention advantages and disadvantages of new technologies; certainly, full-ceramic and hybrid bearings have been also studied. Please consider the following papers: https://doi.org/10.3390/s21175832 ; https://doi.org/10.1007/s12555-021-0167-0

2. Please include the corresponding references and carefully review the following texts "Error! Reference source not found" in the whole document.

3. For a better interpretation of the proposed, it could be interesting to include a step-by-step flow chart.

4. More details in the description of the test ring are required (Section 2.2), if possible, include a representation of the used test bench and please include name and details about the location of sensors; and/or, include different views of the real test bench and include name of the elements in order to identify the different parts as well as sensors.

5. What is the performance of the proposed work in comparison with classical approaches? is the proposal more effective than approaches based on vibrations or stator currents?

6. Please mention in the Conclusion section if this proposal is suitable to be implemented as a part of the Condition Monitoring Based Maintenance in industrial sites? and also, include future work.

The contribution of this proposal must be highlighted 

Author Response

The authors thank you for your revision. With your feedback, we were able to improve our work. You'll find our answers to your comments written below.

In literature review presented in Section 1 presents some of the most relevant research that have been already published, indeed, the proper organization of Section 1 allows to understand challenges that are mandatory to overcome. In this sense, new material technologies have emerged, and it could be interesting to include a brief paragraph to mention advantages and disadvantages of new technologies; certainly, full-ceramic and hybrid bearings have been also studied. Please consider the following papers: https://doi.org/10.3390/s21175832 ; https://doi.org/10.1007/s12555-021-0167-0

Because of the usage of the electric properties of rolling element bearings, hybrid bearings or full ceramic bearings used in applications like electric machinery cannot be observed using the impedance due to the missing electrical conductivity. For these bearings, classic monitoring approaches have to be used and optimized using feature engineering and other techniques

Please include the corresponding references and carefully review the following texts "Error! Reference source not found" in the whole document.

This has been fixed

For a better interpretation of the proposed, it could be interesting to include a step-by-step flow chart.

The flow chart is implemented in the paper.

More details in the description of the test ring are required (Section 2.2), if possible, include a representation of the used test bench and please include name and details about the location of sensors; and/or, include different views of the real test bench and include name of the elements in order to identify the different parts as well as sensors.

We enlarged the description.

What is the performance of the proposed work in comparison with classical approaches? is the proposal more effective than approaches based on vibrations or stator currents?

In further research, a comparison of the impedance signal features to the advanced vibration analysis and motor current analysis is necessary. Other papers could show the possibilities of vibration analysis using e.g. deep feature learning. They are material independent like mentioned in section 1.3, which allows a broader application field. In case of motor current based condition monitoring, additional sensors are not needed, which is an important cost factor. Impedance based condition monitoring is applicable for slow rotating machinery or critical processes and systems. For a higher data quality, disturbance factors have to be identified and analyzed. For system applications, the exact electrical paths through the structure have to be known. First research results at the institute could show, that disturbance factors have a specific behavior, what has also to be investigated further for real applications.

Please mention in the Conclusion section if this proposal is suitable to be implemented as a part of the Condition Monitoring Based Maintenance in industrial sites? and also, include future work.

In addition, the changes of bearing type and scale have to be investigated further to ensure that the impedance is independent from these factors. In this paper, ball bear-ings have been used as test bearings. In the future, roller bearings and bearings with line contact in general have to be researched.

The results presented in this paper show the opportunities of impedance based condition monitoring. As mentioned in section 4.3, the technique can be applied for special use cases, where vibration analysis is not sufficient for condition monitoring. The implementation in industrial gearboxes for their observation is already part of the research at the institute. Indicators could be found that the impedance measurement is able to observe not only rolling element bearings themselves, but also the entire gear-box. In this case, further research about the impedance behavior is needed.

Reviewer 3 Report

In this paper, the impedance signal and its features during the operational time of rolling bearing are studied, in which some problems exist as follows:

1. The clarity of the illustrations in the text is too low with significant distortion, e.g., Fig 7 and Fig 8; please employ high-definition pictures. Additionally, all links to images in the text are reported as errors, please check them carefully.

2. The expression of the whole paper should be embellished since it reads unlike a scientific paper. (eg. “Looking at the correlation coefficients, there are some very high ranked features and a lot of features with a very weak correlation to the normalized accumulated total damage” in line 303)

3. The parts contained within the test bench mentioned in the text should be labeled one by one in Figure 5, otherwise it is pointless to have only an undescribed photo.

4. The horizontal coordinate range in the lower left corner of Figures 10 and Figures 11 does not match "First 60 Minutes of the Test", please check.

5. Some references in Introduction are cited too frequently. Please note that academic ethics issues may be triggered when a paper is over-referenced. Furthermore, please endeavor to refrain from making citations in the results and discussion which should present your own research findings.

6. To guarantee the accuracy of the test results, suggest improving the test protocol to eliminate the interference caused by the disassembly behavior during the test.

Author Response

The authors thank you for your feedback! It helped us to inprove our paper. You'll find our answers to your comments written below.

The clarity of the illustrations in the text is too low with significant distortion, e.g., Fig 7 and Fig 8; please employ high-definition pictures. Additionally, all links to images in the text are reported as errors, please check them carefully.

We fixed the problems addressed here in the paper.

The expression of the whole paper should be embellished since it reads unlike a scientific paper. (eg. “Looking at the correlation coefficients, there are some very high ranked features and a lot of features with a very weak correlation to the normalized accumulated total damage” in line 303)

We improved the expressions and the language in general.

The parts contained within the test bench mentioned in the text should be labeled one by one in Figure 5, otherwise it is pointless to have only an undescribed photo.

We pointed the mentioned components in the figure.

The horizontal coordinate range in the lower left corner of Figures 10 and Figures 11 does not match "First 60 Minutes of the Test", please check.´

The coordinate range is now matching.

Some references in Introduction are cited too frequently. Please note that academic ethics issues may be triggered when a paper is over-referenced. Furthermore, please endeavor to refrain from making citations in the results and discussion which should present your own research findings.

We improved it. We also tryed to avoid references in the results, but for explanation and discussion results from other authors was necessary, which is not addressed in section 1.

To guarantee the accuracy of the test results, suggest improving the test protocol to eliminate the interference caused by the disassembly behavior during the test.
We considered it in the text and will avoid it in future tests.
“This can be achieved by exchanging both test bearings after a detected bearing damage instead of exchanging only the damaged one.”

Round 2

Reviewer 1 Report

  1. The purpose of condition monitoring is to detect the remaining useful life. So, the authors should elaborate on this to enhance the significance of the manuscript (e.g., doi:10.1007/s00170-022-09377-9, doi:10.3390/s16060795).

Need proofreading

Author Response

Dear reviewer,

we thank you again for your revision. We clarified in the text, that RUL prediction is not part of our research yet. Mentioning it in the introduction may led to confusions regarding the aim of our research presented here, so we edited it. Our aim is to identify features in the impedance signal to get information about the rolling bearing condition. So we apologize for not including the references you sent us. The observation of the RUL of machine tools and the tool life prognosis is an interesting research topic, but not part of our research.

As mentioned in our contribution, we will consider the usage of RUL prediction algorithms in further research due to our results.

We edited the language as well.

Kind regards,

the authors

Reviewer 2 Report

The manuscript has been improved and all the comments have been correctly addressed.

There are no additional comments.

Author Response

Thank again for your review!